

# Identification and quantification of flavonoids in 207 cultivated lotus (*Nelumbo nucifera*) and their contribution to different colors

Jing Liu[1], Yuetong Yu[1], Gangqiang Dong[2], Chenyang Hao[1], Yan Liu[1] and Sha Chen[1]

[1] Key Laboratory of Beijing for Identification and Safety Evaluation of Chinese Medicine, Institute of Chinese Materia Medica, China Academy of Chinese Medical Sciences, No. 16, Nanxiaojie, Dongzhimennei, Beijing, China

[2] Amway (China) Botanical R&D Centre, Wuxi, Jiangsu

## ABSTRACT

Sacred lotus (*Nelumbo nucifera*) is a large economic crop, which is also cultivated as a horticultural crop. This study performed a systematic qualitative and quantitative determination of five anthocyanins and 18 non-anthocyanin flavonoids from the petals of 207 lotus cultivars. Among the compounds identified in this study, quercetin 3-*O*-pentose-glucuronide, quercetin 7-*O*-glucoside, laricitrin 3-*O*-hexose, and laricitrin 3-*O*-glucuronide were discovered for the first time in sacred lotus. The relationships between these pigments and petals colors were also evaluated. A decrease in the total content of anthocyanins and increase in the content of myricetin 3-*O*-glucuronide resulted in a lighter flower color. Furthermore, petals were yellow when the content of quercetin 3-*O*-neohesperidoside and myricetin 3-*O*-glucuronide were increased, whereas petals were red when the total anthocyanin content was high and the quercetin 3-*O*-sambubioside content was low. These investigations contribute to the understanding of mechanisms that underlie the development of flower color and provide a solid theoretical basis for the further study of sacred lotus.

Corresponding authors
Yan Liu, yliu1980@icmm.ac.cn
Sha Chen, schen@icmm.ac.cn

## INTRODUCTION

Sacred lotus (*Nelumbo nucifera*) is a large economic crop that is also cultivated as a horticultural crop; its seeds and underground stems are commonly used as vegetables, and its flower has an ornamental value. It has a long cultivated history with a rich variety resource. Based on morphological characteristics (*Guo, 2009*; *Mukherjee et al., 2009*), more than 500 cultivars of *N. nucifera* exist and are native to Asia and Australia, while *N. lutea* has yellow petals and is native to North America. It is acknowledged that sacred lotus petals present different colors, including red, pink, yellow, white, and red/white pied. The lotus cultivars, "Feihong" "Fenhonglingxiao" "Guoqinghong" "Honghuajianlian" "Shaoxinghonglian" and "Yanyangtian", attract widespread attention because of their

bright red, while "Yuwan" "Xueju" "Baijunzixiaolian" and "Baixuegongzhu" are loved by people owing to their pure white color. It is worth mentioning that yellow occupies a special position in lotus flower colors, mainly from "Meizhouhuanglian" and its hybrid descendants. Moreover, anthocyanins are known to be the key factors in the diversity of sacred lotus colors (*Deng et al., 2013*; *Yang et al., 2009*). Because of the presence of anthocyanins and non-anthocyanin flavonoids, lotus exhibit many beneficial biological activities, such as antioxidant, anti-inflammatory, antibiotic, antiallergic and antitumor activities (*Chen et al., 2012*; *Jung et al., 2008*; *Jung et al., 2003*; *Juranić & Žižak, 2005*; *Zhu et al., 2013*).

*Deng et al. (2013)* systematically analyzed the composition and content of anthocyanins, flavones, and flavonols in 108 sacred lotus cultivars with different petals colors. Furthermore, *Chen et al. (2013)* proposed a putative flavonoid biosynthetic pathway in sacred lotus; however, a branch of the suggested pathway remains incomplete. To explore the flower coloration mechanism in sacred lotus, *Deng et al. (2015)* conducted a comparative proteomics analysis of petals from red and white cultivars and found that different methylation intensities on the promotor sequences of the anthocyanin synthase gene may contribute to the diversity of petal colors. In addition, *Sun et al. (2016)* validated that *NnMYB5* is a transcription activator of anthocyanin synthesis and the color difference between red and yellow sacred lotus species may be related to a variation in the *MYB5* gene. These studies have shown that it is pertinent to investigate the mechanism that underlies color formation in sacred lotus and further study is required.

It appears that a correlation between chemical composition and color phenotype may exist in sacred lotus. To further investigate the coloring mechanism of sacred lotus petals, a large number of sacred lotus samples were collected from all over the world, comprising examples of almost all the colors that exist in this species. Based on this collection, we systematically detected, qualified, and quantified the contents of anthocyanins and non-anthocyanin flavonoids in 207 sacred lotus cultivars, and measured the petal color phenotypes using spectrophotometry. In addition, correlations between petal color and the presence of different pigments were analyzed. This work may benefit our understanding of the relationship between the composition of flavonoids and petal color in sacred lotus, while providing a basis for subsequent research on this important plant species.

## MATERIALS & METHODS

### Chemicals and materials

The anthocyanin standard petunidin 3-*O*-glucoside (≥98.0%) and flavonol standards hyperoside, astragalin, and isorhamnetin 3-*O*-glucoside (≥98.0%) were purchased from Chengdu Push Bio-technology Co., Ltd. Acetonitrile and formic acid were obtained from Sigma-Aldrich (St. Louis, MO, USA), which were applied as eluent and eluent additive in ultra high-performance-liquid chromatography (UPLC) and UPLC-mass spectrometry (UPLC-MS). Other analytical grade reagents were purchased from the Beijing Chemistry Factory (Beijing, China). UPLC-grade water was obtained from Watsons water. Millipore membranes (0.22 μm) were acquired from Alltech Scientific (Beijing, China). The samples

were powdered in liquid nitrogen using an analytical mill (IKA A11 basic machine; IKA, Staufen, Germany).

## Plant materials

Petals of 207 sacred lotus cultivars (Table S1) were grown in the United Lotus Germplasm Resource of the Amway Plant Research and Development Center and the Chinese Academy of Traditional Chinese Medicine (China, WuXi, lat. 31°57′, N long, 120°29′) in the same-sized containers (height: 90 cm, diameter: 70 cm), while receiving the same fertilization and disease control treatments. Two days after the bracts emerged, three biological replicates of petals from each cultivar were manually collected, during May and June of 2018 (ambient temperature, 26–30 °C), from three individual plants. The fresh petals were immediately frozen in liquid nitrogen, powdered with an analytical mill (IKA AII basic; IKA, Staufen, Germany), and then stored at −80 °C until later use.

## Color analysis

The fresh petals were compared to the Royal Horticultural Society Color Chart (RHSCC) and sorted into four color groups, including purple-red, red, yellow, and white. The colors of the lotus flowers were measured using a spectrophotometer (NF555, Nippon Denshoku, Japan). For each lotus flower, petals were randomly selected, except for those in the outermost and innermost layers. The selected petals were then measured at a viewing angle of 2° under Illuminant C. And ColorMate software (version 5) was adopted to collect and process the values of $L^{\star}$, $a^{\star}$, $b^{\star}$, $C^{\star}$, and h. The $L^{\star}$ value symbolizes the lightness of the color. With L value increased, the color becomes lighter, from black ($L^{\star} = 0$) to white ($L^{\star} = 100$). In addition, positive and negative $a^{\star}$ values separately represent red and green, while positive and negative $b^{\star}$ values on behalf of yellow and blue, respectively. Two new parameters, chroma $[C^{\star} = (a^{\star 2} + b^{\star 2})^{1/2}]$ and hue angle $[h = \arctan b^{\star}/a^{\star}]$, were derived from $a^{\star}$ and $b^{\star}$. The chroma parameter describes the saturation of the color, while the hue angle value is stepped counterclockwise across a continuously fading hue circle (*Gonnet, 1998*; *Gonnet, 1999*). The co-pigment index (CI) value $[CI = TF/TA]$, which represents the co-pigmentation effect, is calculated from the total content of non-anthocyanin flavonoids (TF) and the total content of anthocyanins (TA). TF and TA will be described in the section "Anthocyanin and non-anthocyanin flavonoid profiles in sacred lotus petals".

## Extraction of anthocyanins and non-anthocyanin flavonoids

The petals were ground into fine powders in liquid nitrogen using an analytical mill (IKA A11 basic machine; IKA, Staufen, Germany). All of the collected samples were extracted according to the method reported by *Deng et al. (2013)* and *Chen et al. (2013)*, with the following modifications: a solvent system comprising methanol, water, and formic acid (70:28:2, v:v:v) was applied in the extraction, and 1 g of sacred lotus petals was extracted with eight mL of extraction buffer and sonicated for 20 min at room temperature. The extracts were centrifuged at 5000× g for 10 min, and the supernatants filtered through a 0.22 μm Millipore filter (Alltech Scientific Corporation, Beijing, China) prior to UPLC analysis.

## HPLC analysis of flavonoids

The analysis of flavonoids was carried out using a Waters H-Class UPLC system consisting of an auto-sampler and quaternary pump arrangement (Waters Corporation, USA) coupled to a UV–vis detector. Compared with previous reports (*Chen et al., 2012*; *Lin & Harnly, 2007*; *Nováková, Matysová & Solich, 2006*), our UPLC method showed higher separation efficiency and resulted in a shorter run-time. A 5 µl aliquot of each sample solution was injected and analyzed on a Waters Xselect $C_{18}$ column (150 mm × 4.6 mm, 3.5 µm, Waters, USA). In the solvent system, eluent A was 10% formic acid in water and eluent B was 0.1% formic acid, added in acetonitrile, as the organic phase. Chromatograms were acquired at 520 nm and 350 nm for anthocyanins and non-anthocyanin flavonoids, respectively. The gradient elution conditions for the separation of the extracted flavonoids were as follows: 0–10 min, 8–15% B; 10–19 min, 15–21% B; 19–22 min, 21% B; 22–23 min, 21–98% B; 23–35 min, 98% B; 35–35.1 min, 98–8% B; 35.1–50 min, 8% B; flow rate, 0.5 mL min$^{-1}$; and temperature, 30 °C.

Anthocyanins and non-anthocyanin flavonoids were quantitatively analyzed with reference to external standards (petunidin 3-*O*-glucoside and hyperoside). The calibration curves showed good linear regression within test concentration ranges, with $R^2 = 0.9972$ and 0.9994, respectively. The limits of detection of the optimized method, calculated as a signal-to-noise ratio of three, were 0.059 µg mL$^{-1}$ and 0.016 µg mL$^{-1}$ for petunidin 3-*O*-glucoside and hyperoside, respectively, while the limits of quantification, with a signal-to-noise ratio of 10, were 0.228 µg mL$^{-1}$ and 0.063 µg mL$^{-1}$, respectively. In addition, the newly developed method provided satisfactory precision and accuracy with overall intra-day and inter-day variations of 0.09–3.41% and 0.66–3.91%, respectively. These results indicated that the optimized UPLC method was stable and suitable for use in the quantitative analysis of flavonoids in sacred lotus petals. In addition, content of compounds **3**, **11**, **19**, and **21** were quantified by comparison with external standards, while compounds **1**, **2**, **4**, and **5** are given in ug/g FW equivalent of petunidin 3-*O*-glucoside. The other non-anthocyanin flavonoids were quantified as hyperoside.

## UPLC-ESI-Q-TOF-MS/MS analysis for determination of flavonoids

Flavonoids in the sacred lotus petal extracts were identified using an Agilent 1290 photodiode array and 6540 triple quad mass time-of-flight (Q-TOF) mass spectrometer, equipped with a dual electrospray ionization (ESI) detector (Agilent, Palo Alto, CA, USA). Nitrogen auxiliary gas was provided. ESI was performed in the negative ionization (NI) mode for both MS and tandem MS (MS/MS) analysis to provide fragmentation information about the molecular weights of the molecules being screened. The ESI source operation parameters were optimized as follows: gas temperature, 350 °C; drying gas, 8 L min$^{-1}$; nebulizer, 45 psig; sheath gas temp, 350 °C; sheath gas flow, 11 L min$^{-1}$; Vcap, 3,500 V; nozzle voltage, 1,500 V; and scan range, m/z 100–1,100 units. A collision energy of 20 eV was used during MS/MS analysis. Purine and HP-0921 were used as internal references in real time and, in NI mode, their m/z ratios were 119.0363 and 1,033.9881, respectively. The MS data, retention times, and UV–vis spectra were used to identify the flavonoids contained in the sacred lotus petals.

## RNA extraction and qRT-PCR analysis

Total RNA was isolated from petals (B88, white petal; A89, yellow petal; B121, red petal) using RNA Easy Fast Plant Tissue Kit purchased from TIANGEN BIOTECH CO., LTD (Beijing, China). Each RNA sample was treated with RNase-free DNase I (TaKaRa) prior to the reverse transcription (RT) reaction to eliminate contaminating genomic DNA. RT-PCR was performed on the basis of the standard instruction of Prime Script RT Reagen Kit with gDNA Eraser (TaKaRa).

As previously described in (*Sun et al., 2016*), the qRT-PCR was carried out using Step One Real-Time PCR system (Applied Biosystems, Foster City, CA, USA). A total reaction volume of 25 uL was applied, containing 10 uL of 2 × TransStart Green PCR Supermix UDG (S602, TRANS), 4 uM of each primer, and about 100 ng of template cDNA. And the amplification condition was follows: incubation at 95 °C for 2 min, denaturation at 95 °C for 5 s, annealing at 60 °C for 10 s, and extension at 72 °C for 10 s, and the process continued for a total of 40 cycles. Besides, action gene of lotus (GenBank ID: EU131153) served as a constitutive control. Target gene relative expression levels were calculated by $2^{-\Delta\Delta Ct}$ comparative threshold cycle (Ct) method, and three biological replicates were conducted. Primer sequences (Table S2) were designed on the whole-genome resequencing data of *N. nucifera* (−30X coverage depth). Three main genes in flavonoid biosynthesis pathway of lotus (DFR (Gene_ID: NW_010729118.1_renew:02005922_02019073), UFGT (Gene_ID: NW_010729304.1_renew:00079217_00080656), OMT (Gene_ID: NW_010729121.1_renew:03748642_03755897) by quantitative reverse transcription −PCR (qRT-PCR) were investigated.

## Statistical analysis

Data were analyzed using SPSS 24.0 for Windows®. The color parameters and pigment contents of petals from 207 cultivars were compared by analysis of variance, combined with Duncan's multiple range tests. Multiple linear regressions (MLR) were established to study the interactions between pigment compositions and color formation.

# RESULTS

## Identification of anthocyanins and non-anthocyanin flavonoids

Flavonoids were identified according to the accurate molecular and fragment ion information obtained using MS and MS/MS, UV–vis spectra, and retention times on the C18 column, as revealed by HPLC and HPLC-MS. Ultimately, five anthocyanins and 18 non-anthocyanin flavonoids were identified (Table 1, Fig. 1A). Flavonoids glycosylated with monosaccharide glycosides show mass spectrometric behavior when using MS with ESI in NI mode. As reported (*Ablajan et al., 2006*), when the abundance of the radical aglycone, annotated as [A −2H] −, is notably higher than that of the aglycone product ion, annotated as [A −H] −, the glycoside is usually linked at the 3-position, and the opposite abundance trend occurs when the conjugation occurs at the 7-position. However, as an exception, only the aglycone product ion can be produced when there is a glucuronic acid group, no matter whether the linked position is 3 or 7. The neutral loss of fragment ions of 146 and 176 mass units, which were produced by the protonated precursor 623.1475

[M-H]- for peak 8, implied that a pentose and a glucuronide were linked at the 3-position. Moreover, the aglycone product ion at m/z 301.0467 [A-H]- in NI mode indicated that the flavonoid aglycone is quercetin, and thus peak 8 was tentatively identified as quercetin 3-*O*-pentose-glucuronide (Qc-3-Pen-Gln) (Fig. 1B). As regard to the peak 9, due to the neutral loss of a fragment ion of 162 mass units, which was produced by the protonated precursor of 463.0910 [M-H]-, and the intensive showing of an aglycone product ion at m/z 301.0362 [A-H]- indicated that the linked position is 7 (Fig. 1C), thus peak 9 was temporarily identified as quercetin 7-*O*-glucoside (Qc-7-Glu). Furthermore, based on the MS/MS spectra data, both peak 13 and peak 14 were tentatively assigned as laricitrin monosaccharide, which has been reported in grapes (*Jin et al., 2009*). The radical aglycone ion at m/z 330.0482 [A-2H]- and the corresponding ion at m/z 315.0228 [A-2H-CH$_3$]-, with a fragment ion of 163 mass units, demonstrated that a hexose substituent was linked at the 3-position (Fig. 1D). Hence, peak 13 was tentatively identified as laricitrin 3-*O*-hexose (Lar-3-hex); and further work is required to identify the nature of this hexose-based compound. Meanwhile, according to the data acquired for the glycone product ion at m/z 331.0581 [A-H]-, and the protonated precursor of 507.0999 [M-H]-, the loss of a fragment ion of 176 mass units manifested that a glucuronic acid glycoside was conjugated at the 3-position. In addition to the corresponding ion at m/z 316.0342 [A-H-CH$_3$]- (Fig. 1E), peak 14 was assigned as laricitrin 3-*O*-glucuronide (Lar-3-Gln).

The chromatographic and MS data for the anthocyanins and non-anthocyanin flavonoids separated and identified from the sacred lotus petals are listed in Table 1. The data from the MS analysis in NI mode provided valuable information, including molecular weights and information about the presence of aglycones and sugars, with their linkage positions. Peaks 1-5 were identified as the following anthocyanins: delphinidin 3-*O*-glucoside (Dp-3-Glu, **1**), cyanidin 3-*O*-glucoside (Cy-3-Glu, **2**), petunidin 3-*O*-glucoside (Pt-3-Glu, **3**), peonidin 3-*O*-glucoside (Pn-3-Glu, **4**), and malvidin 3-*O*-glucoside (Mv-3-Glu, **5**), as previously reported (*Yang et al., 2009*). Peaks 6-23 were identified as non-anthocyanin flavonoids: myricetin 3-*O*-glucuronide (Myr-3-Gln, **6**) (*Deng et al., 2013*), quercetin 3-*O*-xylopyranosyl-(1→2)-glucopyranoside (quercetin 3-*O*-sambubioside/Qc-3-Sam, **7**) (*Deng et al., 2009*), quercetin 3-*O*-pentose-glucuronide (Qc-3-Pen-Gln, **8**), quercetin 7-*O*-glucoside (Qc-7-Glu, **9**), quercetin 3-*O*-rhamnopyranosyl-(1→2)-galactopyranoside (quercetin 3-*O*-neohesperidoside/Qc-3-Neo, **10**) (*Li et al., 2014b*), quercetin 3-*O*-galactoside (Qc-3-Gal/Hyperoside, **11**) (*Suzuki et al., 2008*), quercetin 3-*O*-glucuronide (Qc-3-Gln, **12**) (*Yang et al., 2009*), laricitrin 3-*O*-hexose (Lar-3-hex, **13**), laricitrin 3-*O*-glucuronide (Lar-3-Gln, **14**), kaempferol 3-*O*-rhamnopyranosyl-(1→2)-glucopyranoside (kaempferol 3-*O*-neohesperidoside/Kae-3-Neo, **15**) (*Lim et al., 2006*; *Sharma et al., 2017*), kaempferol 3-*O*-galactoside (Kae-3-Gal, **16**) (*Jung et al., 2003*), kaempferol 3-*O*-rhamnopyranosyl-(1→6)-glucopyranoside (kaempferol 3-*O*-rutinoside/Kae-3-Rut, **17**) ;  (*Hyun et al., 2006*; *Sharma et al., 2017*), isorhamnetin 3-*O*-rutinoside (Iso-3-Rut, **18**) (*Yang et al., 2009*), kaempferol 3-*O*-glucoside (Kae-3-Glu/astragalin, **19**) (*Chen et al., 2012*; *Yang et al., 2009*), syringetin 3-*O*-hexose (Syr-3-Hex, **20**) (*Guo, 2009*), isorhamnetin 3-*O*-glucoside (Iso-3-Glu, **21**) (*Sharma et al., 2017*),

Liu et al. (2022), *PeerJ Analytical Chemistry*, DOI 10.7717/peerj-achem.22

**Table 1  Identification of flavonoids in petals of lotus.**

| Peak No. | Rt (min)[a] | λ max(nm) | Parent ion(m/z) (measured value) | Molecular fomular | Parent ion(m/z) (calculated value) | Error (ppm) | Fragmentation profile(m/z) (Relative abundance %) | Identification | References |
|---|---|---|---|---|---|---|---|---|---|
| 1 | 4.765 | 523.9, 272.2 | 463.0904[M-2H]⁻ | $C_{21}H_{21}O_{12}^{+}$ | 463.0882 | −4.75 | 301.0376(100),300.0358(95.28) | Delphinidin 3-O-glucoside(Dp-3-Glu) | *Yang et al. (2009)* |
| 2 | 5.959 | 519.0, 279.3 | 447.0951[M-2H]⁻ | $C_{21}H_{21}O_{11}^{+}$ | 447.0933 | −4.03 | 285.0448(100),284.0372(57.11) | cyanidin 3-O-glucoside(Cy-3-Glu) | *Yang et al. (2009)* |
| 3 | 6.770 | 523.9, 279.3 | 477.1046[M-2H]⁻ | $C_{22}H_{23}O_{12}^{+}$ | 477.1038 | −1.68 | 315.0547(100),314.0485(87.66) | petunidin 3-O-glucoside(Pt-3-Glu)[b] | *Yang et al. (2009)* |
| 4 | 8.418 | 517.8, 279.3 | 461.1099[M-2H]⁻ | $C_{22}H_{23}O_{11}^{+}$ | 461.1089 | −2.17 | 299.0597(100),298.0519(55.27) | peonidin 3-O-glucoside(Pn-3-Glu) | *Yang et al. (2009)* |
| 5 | 9.271 | 527.6, 277.0 | 491.1208[M-2H]⁻ | $C_{23}H_{25}O_{12}^{+}$ | 491.1195 | −2.65 | 329.0711(100),328.0644(69.32) | malvidin 3-O-glucoside(Mv-3-Glu) | *Yang et al. (2009)* |
| 6 | 11.591 | 355.4, 264.7 | 493.0640[M-H]⁻ | $C_{21}H_{18}O_{14}$ | 493.0624 | −3.25 | 317.0418(100),318.0451(24.47) | myricetin 3-O-glucuronide(Myr-3-Gln) | *Deng et al. (2013)* |
| 7 | 12.268 | 356.6, 252.0 | 595.1309[M-H]⁻ | $C_{26}H_{28}O_{16}$ | 595.1305 | −0.67 | 300.0380(100),301.0433(21.23) | quercetin 3-O-sambubioside (Qc-3-Sam/Qc-3-Xyl-Glu) | *Deng et al. (2009)* |
| 8 | 12.606 | 356.6, 253.2 | 623.1475[M-H]⁻ | $C_{27}H_{27}O_{17}$ | 623.1254 | −11.72 | 301.0467(100),302.0502(13.95),446.7323(2.46) | quercetin 3-O-pentose-glucuronide (Qc-3-Pen-Gln)[c] | *Ablajan et al. (2006)* |
| 9 | 14.349 | 319.9, 252.0 | 463.0910[M-H]⁻ | $C_{21}H_{20}O_{12}$ | 463.0882 | −6.05 | 301.0362(100),300.0251(20.05) | quercetin 7-O-glucoside(Qc-7-Glu)[c] | *Ablajan et al. (2006)* |
| 10 | 14.687 | 352.9, 268.6 | 609.1463[M-H]⁻ | $C_{27}H_{30}O_{16}$ | 609.1461 | −0.33 | 609.1461(100),300.0276(50.58),301.0325(36.65) | quercetin 3-O-neohesperidoside(Qc-3-Neo) | *Li et al. (2014b)* |
| 11 | 15.399 | 354.1, 254.4 | 463.0900[M-H]⁻ | $C_{21}H_{20}O_{12}$ | 463.0882 | −3.89 | 300.0274(100),301.0343(64.38) | quercetin 3-O-galactoside(Qc-3-Gal/hyperoside)[b] | *Suzuki et al. (2008)* |
| 12 | 15.913 | 352.9, 255.5 | 477.0690[M-H]⁻ | $C_{21}H_{18}O_{13}$ | 477.0675 | −3.14 | 301.0369(100),302.0394(23.85) | quercetin 3-O-glucuronide (Qc-3-Gln) | *Yang et al. (2009)* |
| 13 | 16.377 | 356.5, 252.0 | 493.0995[M-H]⁻ | $C_{22}H_{22}O_{13}$ | 493.0988 | −1.42 | 330.0482(100),331.0518(56.74),315.0228(19.51) | laricitrin 3-O-hexose(Lar-3-Hex)[c] | *Jin et al. (2009)* |
| 14 | 16.663 | 359.0, 252.0 | 507.0788[M-H]⁻ | $C_{22}H_{20}O_{14}$ | 507.078 | −1.58 | 331.0581(100),332.0569(22.80),316.0341(8.97) | laricitrin 3-O-glucuronide(Lar-3-Gln)[c] | *Jin et al. (2009)* |
| 15 | 17.263 | 343.5, 252.0 | 593.1516[M-H]⁻ | $C_{27}H_{30}O_{15}$ | 593.1512 | −0.67 | 593.1740(100),284.0420(76.83), 285.0471(42.92) | Kaempferol 3-O-neohesperidoside (Kae-3-Neo/Kae-3-Rha-Glu) | *Lim et al. (2006)*; *Sharma et al. (2017)* |
| 16 | 18.374 | 343.5, 265.1 | 447.0952[M-H]⁻ | $C_{21}H_{20}O_{11}$ | 447.0933 | −4.25 | 284.0428(100),284.0492(45.53) | kaempferol 3-O-galactoside(Kae-3-Gal) | *Jung et al. (2003)* |
| 17 | 18.870 | 343.5, 253.2 | 593.1516[M-H]⁻ | $C_{27}H_{30}O_{15}$ | 593.1512 | −0.67 | 285.0500(100),284.042(52.21) | kaempferol 3-O-rutinoside(Kae-3-Rut) | *Hyun et al. (2006)*; *Sharma et al. (2017)* |
| 18 | 19.259 | 352.9, 253.2 | 623.1612[M-H]⁻ | $C_{28}H_{32}O_{16}$ | 623.1618 | 0.96 | 623.1839(100),315.0625(52.24),314.0563(28.12) | isorhamnetin 3-O-rutinoside (Iso-3-Rut) | *Yang et al. (2009)* |
| 19 | 19.770 | 347.0, 265.1 | 447.0951[M-H]⁻ | $C_{21}H_{20}O_{11}$ | 447.0933 | −4.03 | 284.0324(100),285.0499(60.54) | kaempferol 3-O-glucoside(Kae-3-Glu/astragalin)[b] | *Chen et al. (2012)*; *Yang et al. (2009)* |
| 20 | 20.536 | 356.6, 253.2 | 507.1157[M-H]⁻ | $C_{23}H_{24}O_{13}$ | 507.1144 | −2.56 | 344.0694(100),345.0747(39.53) | syringetin 3-O-hexose(Syr-3-Hex) | *Guo (2009)* |
| 21 | 20.821 | 356.6, 253.2 | 477.1051[M-H]⁻ | $C_{22}H_{22}O_{12}$ | 477.1038 | −2.72 | 314.0554(100),315.0595(27.71) | isorhamnetin 3-O-glucoside(Iso-3-Glu)[b] | *Sharma et al. (2017)* |
| 22 | 21.377 | 351.7, 253.2 | 491.0845[M-H]⁻ | $C_{22}H_{20}O_{13}$ | 491.0831 | −2.85 | 315.0522(100),316.0567(28.44) | isorhamnetin 3-O-glucuronide (Iso-3-Gln) | *Chen et al. (2012)* |
| 23 | 21.617 | 357.8, 252.0 | 521.0960[M-H]⁻ | $C_{23}H_{22}O_{14}$ | 521.0937 | −4.41 | 345.0607(100),346.0639(29.29) | syringetin 3-O-glucuronide(Syr-3-Gln) | *Li et al. (2014b)* |

**Notes.**

[a]Rt: retention time on $C_{18}$ column.

[b]Compounds identified by standards.

[c]Compounds identified for the first in sacred lotus.

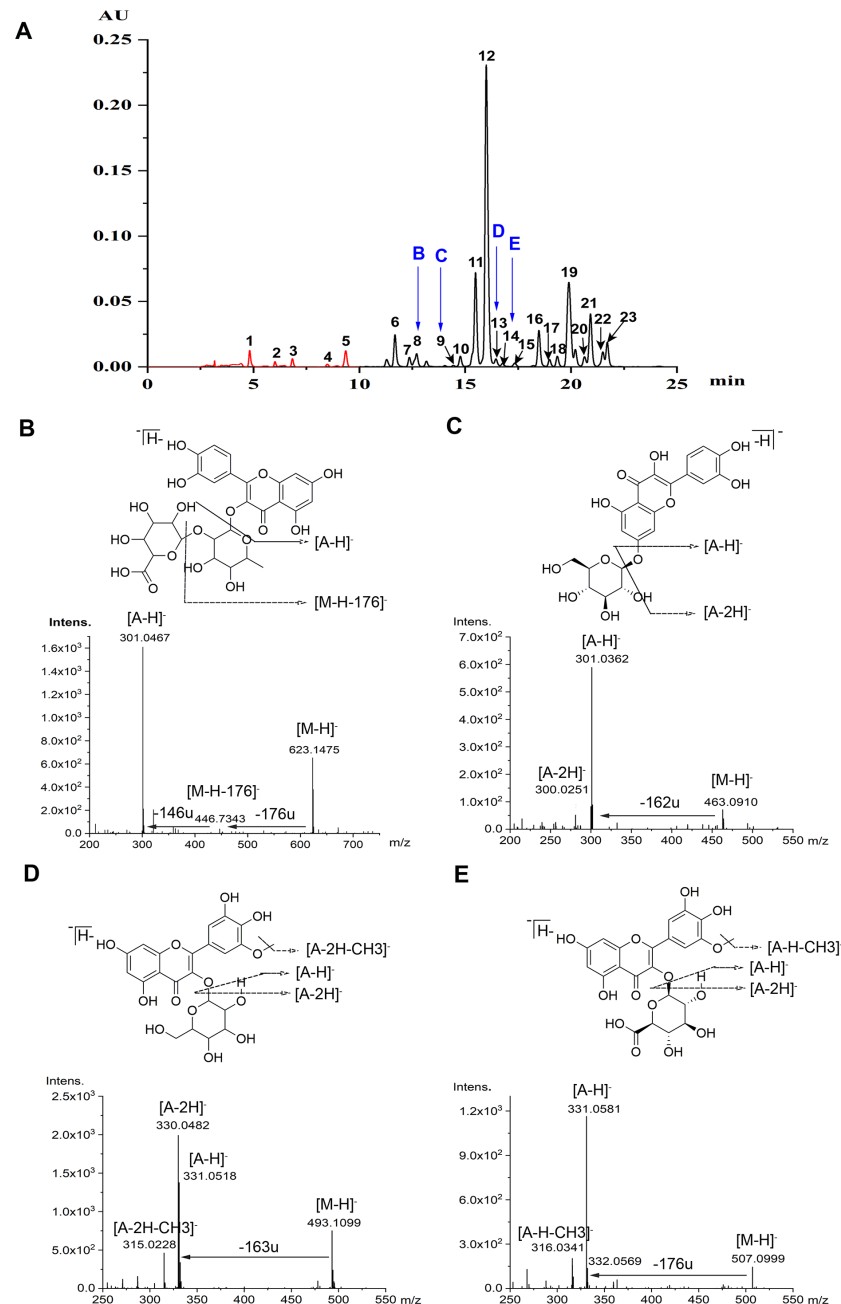

**Figure 1 Chromatograms and MS spectrum of flavonoids detected in lotus petals.** (A) HPLC chromatograms of anthocyanins at 520 nm (peaks 1-5) and of non-anthocyanin flavonoids at 350 nm (peaks 6-23). Peak numbers in this figure correspond to compound numbers in Table 1. These data were obtained from the mixture of three cultivars: "Qiaoshou-I" (71), "Jinlingningcui" (151), and "Silian13-I" (178). (B-D) Structures of four flavonoids found in lotus petals. MS/MS spectra (in NI mode) of quercetin 3-*O*-pentose-glucuronide (B, 8), quercetin 7-*O*-glucoside (C, 9), laricitrin 3-*O*-hexose (D, 13), and laricitrin 3-*O*-glucuronide (E, 14), and produced by each precursor.

isorhamnetin 3-*O*-glucuronide (Iso-3-Gln, **22**) (*Chen et al., 2012*), and syringetin 3-*O*-glucuronide (Syr-3-Gln, **23**) (*Li et al., 2014b*). These 18 non-anthocyanin flavonoids were classified into six groups: quercetin (Qc), kaempferol (Kae), isorhamnetin (Iso), myricetin (Myr), syringetin (Syr), and laricitrin (Lar), based on the aglycones they contain.

In this study, the four non-anthocyanin flavonoids discussed above (Fig. 1), including Qc-3-Pen-Gln (**8**), Qc-7-Glu (**9**), Lar-3-Hex (**13**) and Lar-3-Gln (**14**), were discovered for the first time in sacred lotus petals using the newly developed UPLC-DAD-ESI-Q-TOF-MS/MS technique. Therefore, our study has further refined the research carried out by *Chen et al. (2013)* and *Deng et al. (2013)*.

## Anthocyanin and non-anthocyanin flavonoid profiles in sacred lotus petals

By comparison with the RHSCC, the 207 sacred lotus cultivars were grouped into four color groups: purple-red, red, yellow, and white. Although there were five different anthocyanins and 18 non-anthocyanin flavonoids detected in the flower petals, the compositions and total contents varied significantly among different cultivars. The most abundant anthocyanin was Mv-3-Glu (**5**), which accounted for 50% of the compounds in the purple-red group and 47% of those in the red group. This was followed by Dp-3-Glu (**1**), which accounted for 19% of the compounds in the purple-red group and 17% of those in the red group. As for the non-anthocyanin flavonoids, quercetin-, kaempferol-, and isorhamnetin-derivatives were the dominant non-anthocyanin flavonoids in all of the four color groups; however, quercetin derivatives were the most abundant (Fig. S1). Of note, in yellow cultivars, quercetin derivatives were seen to be the most important flavonoids, as these were present in significantly higher amounts (up to 64%) than the other five flavone aglycone derivatives, suggesting a link between these compounds and the yellow petal color. Additionally, kaempferol and isorhamnetin derivatives were the second and third most abundant complexes in yellow petals, accounting for 16% and 9% of the TF, respectively. Conversely, quercetin and kaempferol derivatives showed equal importance in their contribution to the TF of the purple-red, red, and white color groups, making up approximately 85% of the TF (Table 2). Among all the cultivars tested, the highest TA was detected in cultivars of the purple-red group, with a mean content of 464.47 ug/g fresh weight (FW), followed by the red group, with an average content of 171.74 ug/g FW. Generally, the petals containing more anthocyanins displayed darker colors; for example, the purple-red cultivar 'Cuifuhongya' (27) had the highest mean anthocyanin content (1,133.61 ug/g FW) among all the cultivars. In terms of germplasm assessment, some cultivars, such as 'Qiuwanluoshan' (5), 'Ti-13' (74), and 'Ti-13-I' (9), which contain very high contents of Mv-3-Glu (**5**) and relatively higher TA, may be ideal candidates for breeding purple-red flowers and studying the anthocyanin biosynthesis pathway in sacred lotus. What's more, the TF was the highest in yellow petals, with an average content of 3,517.93 ug/g FW. The yellow petal cultivar 'Jintaiyang' (165) contained the highest TF, with a concentration of 7,149.35 ug/g FW, and would be a candidate cultivar for the study of the coloring mechanism in yellow sacred lotus (Fig. 2).

**Table 2  Average content (ug g⁻¹ fresh weight) of different pigment types in different colors of lotus.**

| Type | Compound | Purple-red | Red | Yellow | White |
|---|---|---|---|---|---|
| Anthocyanins | Dp-3-Glu | 88.93 ± 6.37a[#] | 28.86 ± 4.54b | 0.31 ± 0.31c | 1.74 ± 0.49c |
| | Cy-3-Glu | 38.73 ± 2.19a | 22.00 ± 3.29b | 0.62 ± 0.43c | 1.14 ± 0.40c |
| | Pt-3-Glu | 61.13 ± 2.90a | 19.66 ± 1.85b | 0.58 ± 0.40c | 1.84 ± 0.48c |
| | Pn-3-Glu | 44.05 ± 1.88a | 22.16 ± 1.94b | 0.71 ± 0.50c | 1.28 ± 0.46c |
| | Mv-3-Glu | 231.64 ± 10.27a | 79.06 ± 10.27b | 1.46 ± 0.68c | 4.36 ± 0.92c |
| | TA[*] | 464.47 ± 19.29a | 171.74 ± 19.41b | 3.67 ± 2.10c | 10.36 ± 2.44c |
| Non-anthocyanin flavonoids | Quercetin derivatives | 762.17 ± 33.76c | 1283.71 ± 205.09b | 2254.58 ± 229.40a | 950.51 ± 104.46c |
| | Kaempferol derivatives | 732.82 ± 42.43a | 749.44 ± 80.60a | 566.21 ± 69.88a | 731.88 ± 55.60a |
| | Isorhamnetin derivatives | 129.55 ± 5.89b | 173.74 ± 22.86b | 319.84 ± 43.77a | 194.34 ± 14.45ab |
| | Syringetin derivatives | 18.72 ± 0.76b | 21.98 ± 2.68b | 82.99 ± 10.97a | 49.18 ± 5.52a |
| | Myricetin derivatives | 79.62 ± 3.70b | 92.44 ± 11.79b | 217.35 ± 37.52a | 80.96 ± 9.47b |
| | Laricitrin derivatives | 36.29 ± 2.29b | 39.72 ± 4.48b | 76.95 ± 9.29a | 39.30 ± 2.66b |
| | TF[*] | 1759.17 ± 68.07c | 2361.02 ± 269.12b | 3517.93 ± 289.08a | 2046.18 ± 133.87bc |

**Notes.**

[*]TA, the total content of the five anthocyanins; TF, the total content of the six non-anthocyanin flavonoid derivatives.

[#]Each value presents the mean ± standard deviation (SD) of three independent replicates. Different letters represent significant ($P < 0.05$) differences between means according to analysis of variance (ANOVA) combined with Duncan's multiple range test. And the differences are within the row.

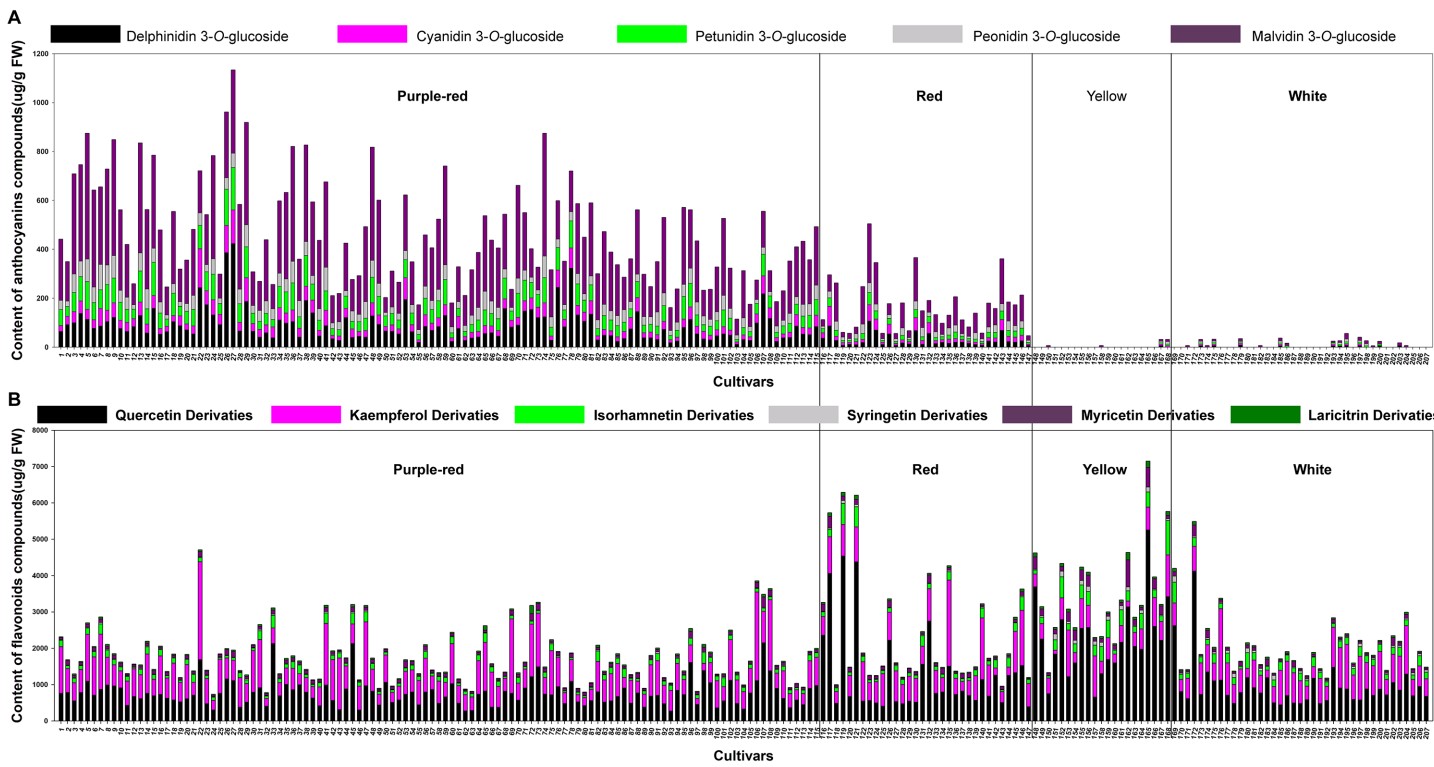

**Figure 2  The average content (ug g⁻¹. fresh weight) of each anthocyanin compound (A) and each non-anthocyanin flavonoid compound (B) in 207 lotus cultivars.** The numbers (*x*-axis) in the figure represent sample numbers, which correspond to the cultivars in Table S1.

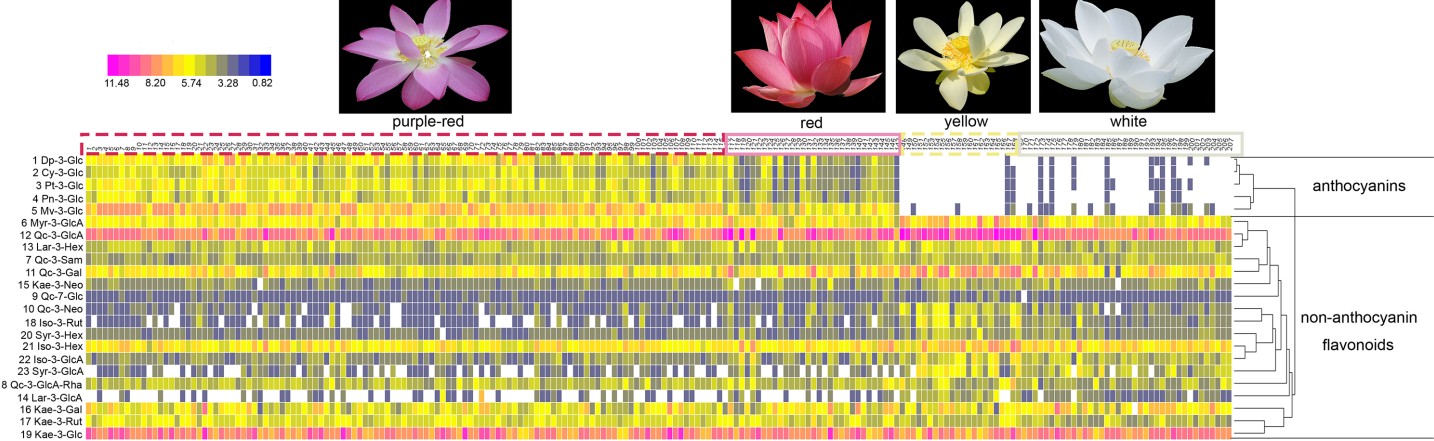

**Figure 3** **A heat map of the individual anthocyanin and non-anthocyanin flavonoid contents.** The row values represent the $Z$-score normalization of the concentration of identified flavonoids and the column represents the tested samples. The numbers on the top line correspond to cultivar numbers in Table S1. Red represents relatively high concentration and blue represents relatively low concentration of the identified flavonoids in petals of each lotus cultivar.

What's more, significant differences were observed in the contents of anthocyanins and non-anthocyanin flavonoids among cultivars of differing color. In order to visualize these differences, individual anthocyanin and non-anthocyanin flavonoid contents were normalized using the $Z$-score and expressed as a heat map (Fig. 3). In purple-red cultivars, Mv-3-Glu (**5**) and Dp-3-Glu (**1**) were the two major anthocyanins, while Qc-3-Gln (**12**) and Kae-3-Glu (**19**) were the two dominant non-anthocyanin flavonoids. The red cultivars exhibited similar profiles, with Mv-3-Glu (**5**) being the most highly concentrated anthocyanin, with Qc-3-Gln (**12**) and Kae-3-Glu (**19**) the most abundant non-anthocyanin flavonoids. The yellow and white cultivars, however, demonstrated a deficiency of anthocyanins and an abundance of non-anthocyanin flavonoids. In yellow cultivars, the concentration of Qc-3-Gln (**12**) was found to be the greatest among the 18 non-anthocyanin flavonoids, which may contribute to the yellow color. Furthermore, the concentrations of Myr-3-Gln (**6**), Qc-3-Neo (**10**), Qc-3-Gal (**11**), Iso-3-Glu (**21**), Iso-3-Gln (**22**), Syr-3-Gln (**23**), Syr-3-Hex (**20**), and Iso-3-Rut (**18**) showed varying degrees of increase, in association with purple-red and red colored cultivars. In white cultivars, the TF exhibited similar trends to those of the yellow cultivars, but at relatively lower levels. Differences in the distribution of the contents of these ingredients suggested that the color of the sacred lotus may be related to a single anthocyanin or non-anthocyanin flavonoid.

## Relationships between petal color, anthocyanins, and non-anthocyanin flavonoids

Researchers have reported that flavones and flavonols are responsible for flower color (*Li et al., 2008*), but the relationships between these factors are unknown in sacred lotus. In maize, non-anthocyanin flavonoids are considered to be co-pigments, alongside anthocyanins (*Stafford, 1998*). Hence the co-pigmentation index is an important indicator of the co-pigmentation effect, which occurs, in the main, when CI >5 (*Asen, Stewart & Norris,*

*1971*; *He et al., 2011*). According to the formula CI = TF/TA, we found that, in the most purple-red sacred lotus petals, CI was <5, while in the other three color groups, CI tended to be >5, indicating that non-anthocyanin flavonoids had a significant effect on the color of sacred lotus petals, especially when the anthocyanin content was low. This result was in line with the suggestion that co-pigmentation between anthocyanins and non-anthocyanin flavonoids may result in distinct petal colors (*Li et al., 2011*; *Zhang et al., 2011b*; *Zhu et al., 2012*).

To analyze the relationship between petal color and pigment content in sacred lotus, Pearson's correlation coefficients were calculated among color parameters, anthocyanin contents, and non-anthocyanin flavonoids contents, and displayed as a heat map (Fig. 4). Strong correlations were seen. For example, L* values were significantly negatively correlated with a*, C*, and h ($P < 0.01$), and significantly positively correlated with b* and CI ($P < 0.01$). The individual anthocyanin contents of the five groups and TA demonstrated significant negative correlations with L*, b*, and CI ($P < 0.01$), and positive correlations with a*, C*, and h ($P < 0.01$). In addition, TF, and most of the individual non-anthocyanin flavonoid contents, were negatively correlated with a*, C*, and h, but positively correlated with L*, b*, and CI. Moreover, Qc-3-Neo (**10**), Qc-3-Gal (**11**), Kae-3-Rut (**17**), Iso-3-Rut (**18**), Syr-3-Hex (**20**), Iso-3-Gln (**22**), and Syr-3-Gln (**23**) were significantly correlated with all of the six color parameters (L*, a*, b*, C*, h, and CI), while Qc-3-Pen-Gln (**8**) and Kae-3-Glu (**19**) had no apparent correlations with most of these color parameters. Correlations among anthocyanin and non-anthocyanin flavonoid metabolism were also evaluated. Strong positive correlations were found between different anthocyanins ($P < 0.01$) and between different non-anthocyanin flavonoids. However, significant negative correlations were observed between anthocyanins and most non-anthocyanin flavonoids ($P < 0.05$), as was found for TA and TF. The correlation analysis indicated that many pigments influence sacred lotus petal color. Thus, MLR analysis was used to estimate the type of pigment that dramatically affects petal color. Color parameters, including L*, a*, and b*, were chosen as dependent variables, and 25 indexes, containing 23 various pigment components, plus TA and TF, were selected as independent variables. To study the interactions between these pigment compositions and color formation, regression equations were established. Significant statistical results were acquired as follows:

$$L* = 76.083 - 0.028TA + 0.022Myr\text{-}3\text{-}GlcA(\mathbf{6})(R^2 = 0.408, P = 3.008E - 24)$$

$$a* = 15.783 + 0.052TA - 0.575Qc\text{-}3\text{-}Sam(\mathbf{7}) + 0.233Kae\text{-}3\text{-}Rut(\mathbf{17}) - 0.003TF$$
$$(R^2 = 0.630, P = 1.079E - 42)$$

$$b* = 0.719 + 0.219Qc\text{-}3\text{-}Neo(\mathbf{10}) + 0.033Myr\text{-}3\text{-}GlcA(\mathbf{6}) + 0.121Syr\text{-}3\text{-}GlcA(23)$$
$$-0.196Pn\text{-}3\text{-}Glc(\mathbf{4}) - 0.325Syr\text{-}3\text{-}Hex(\mathbf{20})(R^2 = 0.570, P = 1.006E - 35)$$

The MLR analysis showed that there are many factors affecting petal color, including TA, TF and the levels of Myr-3-GlcA (**6**), Qc-3-Sam (**7**), Kae-3-Rut (**17**), Qc-3-Neo (**10**), Myr-3-Gln (**6**), Syr-3-GlcA (**23**), Pn-3-Glc (**4**) and Syr-3-Hex (**20**), among which, TA was

the major factor, with positive effects on the values of a*, but negative effects on the value of L*. Myr-3-GlcA (**6**) was another important factor that exhibited positive effects on the L* value. TA was found to be the primary factor positively influencing the a* value, whereas Qc-3-Sam (**7**) negatively affected the a* value. Furthermore, Qc-3-Neo (**10**), Myr-3-Gln (**6**) and Syr-3-GlcA (**23**) had positive effects on value of b*. Based on these findings, an increase in TA was determined to push up the values of a*, but lower the value of L*, indicating that the flower colors would become red and darker. The L* value indicates that with less TA and higher Myr-3-GlcA (**6**) contents, flowers are lighter or white in color. Parameter a* suggests that higher TA and Kae-3-Rut (**17**) contents, with lower Qc-3-Sam (**7**) contents and TF, lead to a deeper red flower color, whereas b* indicates that higher Qc-3-Neo (**10**), Myr-3-Gln (**6**) and Syr-3-GlcA (**23**) levels deepen the yellow color of petals. In summary, increasing the content of Myr-3-GlcA (**6**) and decreasing the TA results in a lighter flower color, whereas a rise in Qc-3-Neo (**10**) and Myr-3-GlcA (**6**) turns petals yellow, while a higher TA and Kae-3-Rut (**17**) contents and lower Qc-3-Sam (**7**) contents turns flowers red.

## Putative flavonoid biosynthesis pathway of lotus

In our study, 5 anthocyanins and 18 non-anthocyanin flavonoids were simultaneously detected, qualified and quantified in 207 sacred lotus cultivars, among which, four components were discovered for the first time in sacred lotus petals. Combined with these newly detected compounds in sacred lotus petals and previous study (*Chen et al., 2013*; *Li et al., 2014b*), the sacred lotus biosynthetic pathway for the detected anthocyanins and non-anthocyanin flavonoids was deduced in depth (Fig. 5). The precursors of flavonoid biosynthesis are 4-coumaroyl-CoA and Malonyl-CoA, which are condensed to form Naringenin under the catalysis of chalcone synthase (CHS) and chalcone isomerase (CHI). Then, with the help of flavonoid 3-hydroxylase (F3H), flavonoid 3′-hydroxylase (F3′H) and flavonoid 3′5′-hydroxylase (F3′5′H), dihydrokaempferol, dihydroquercetin and dihydromyricetin were synthesized, which were the most essential precursor compounds used to synthesize the corresponding anthocyanins and non-anthocyanin flavonoids. As the biosynthesis of flavonols is closely concerning to that of anthocyanins (*Jeong et al., 2006*), the pathway then divided into five sub-pathway. Flavonol synthase (FLS) played a decisive role for producing aglycones of non-anthocyanin flavonoids, while dihydroflavonol reductase (DFR) determined the generation of anthocyanins. The main difference is sub-pathway 3, for the presence of flavonol kaempferol synthesized from dihydrokaempferol, while the anthocyanin pelargonidin is lacking in sacred lotus (*Li et al., 2014b*). Finally, both anthocyanins and non-anthocyanin flavonoids, with the assistance of enzyme UDP flavonoid glycosyltransferase (UFGT), *O*-methyltransferase (OMT) and other enzymes, performed different structural modification at the linkage position. Especially, glycosylation is a key mechanism to coordinate the bioactivity, metabolism and location of small molecules in living cells (*Pfeiffer & Hegedűs, 2011*). As shown in our study, it may be much simpler to glycosylate at 3-*O*-position in sacred lotus petals, with the exception of quercetin. The compound Qc-7-Glu (**9**), glycosylated in the 7-*O*-position, was detected for the first time in sacred lotus. More importantly, the first discovery of laricitrin derivatives

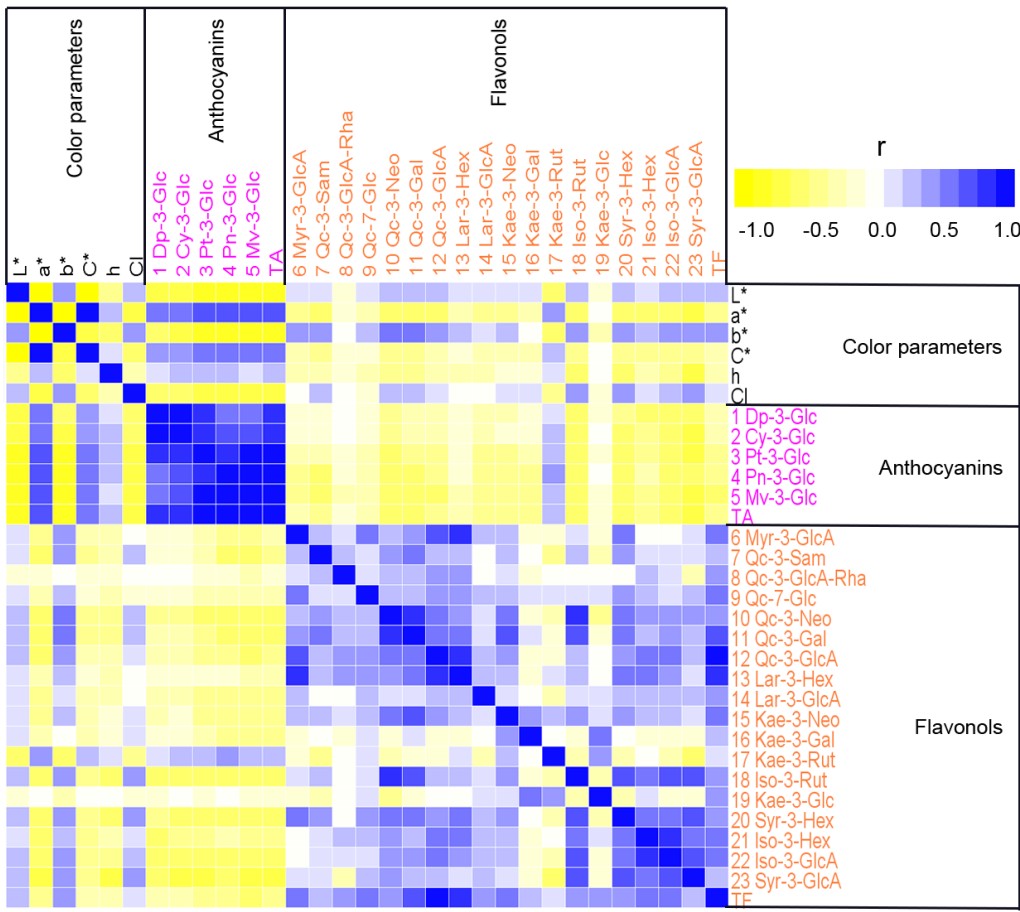

**Figure 4** **A heat map of correlation matrix of color parameters and 23 compounds from petals of 207 lotus cultivars.** Each square indicates Pearson's correlation coefficient for a pair of data, and the intensity of red and green colors in the heat map indicates the level of positive and negative correlation, respectively.

supplemented the sub-pathway 5, which was validated by the results of the large data resources and chemical technologies.

To further valid the flavonoid biosynthetic pathway we proposed, total RNA was isolated from three lotus cultivars with different petal colors, and qRT-PCR was conducted to observe these gene expressions (DFR, OMT, UFGT) in lotus petals with different colors (Fig. 6). The qRT-PCR results showed that the expressions of DFR and UFGT in red petals was significantly higher than those of yellow and white petals, indicating the higher anthocyanin content in red petal cultivars. In addition, the expression levels of OMT genes in yellow petals were significantly higher than those in red and white petals, suggesting the higher non-anthocyanin flavonoids contents in yellow petal cultivars. The qRT-PCR results further validated the putative flavonoid biosynthetic pathway.

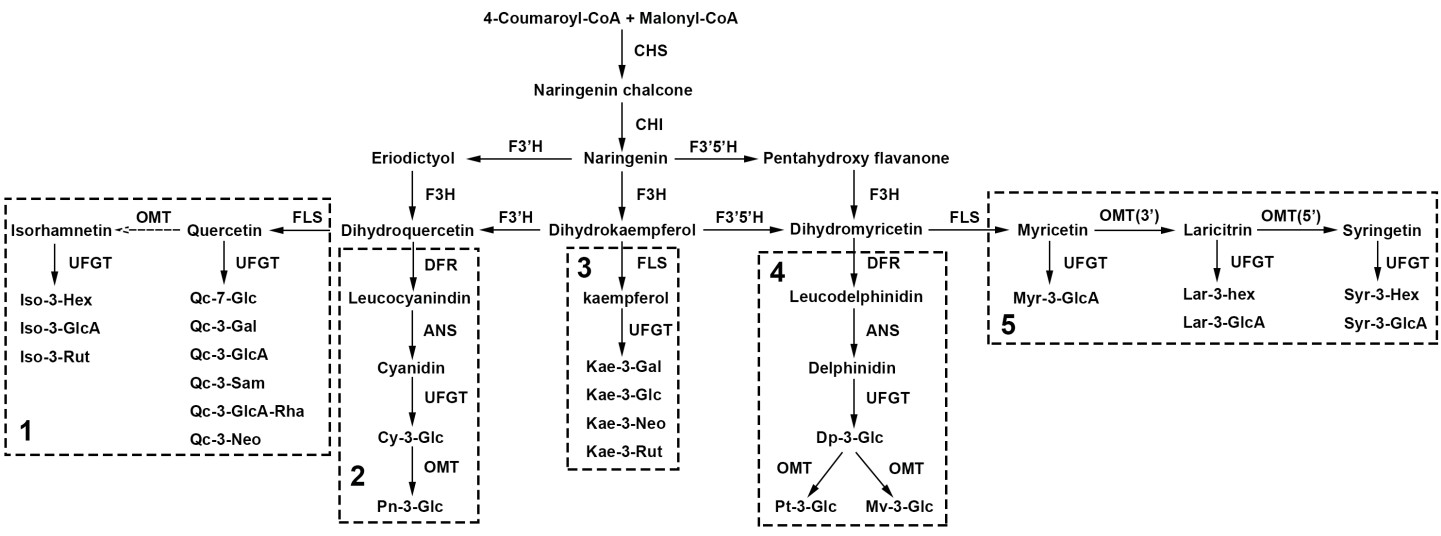

**Figure 5** **The deduced biosynthesis pathway of flavonoids in lotus petals.** CoA, acethl coenzyme A; CHS, chalcone synthase; CHI, chalcone isomerase; F3H, flavonoid 3-hydroxylase; F3′H, flavonoid 3′-hydroxylase; F3′5′H, flavonoid 3′5′-hydroxylase; DFR, dihydroflavonol reductase; ANS, anthocyanidin synthase; FLS, flavonol synthase; UFGT, UDP flavonoid glycosyltransferase; OMT, O-methyltransferase.

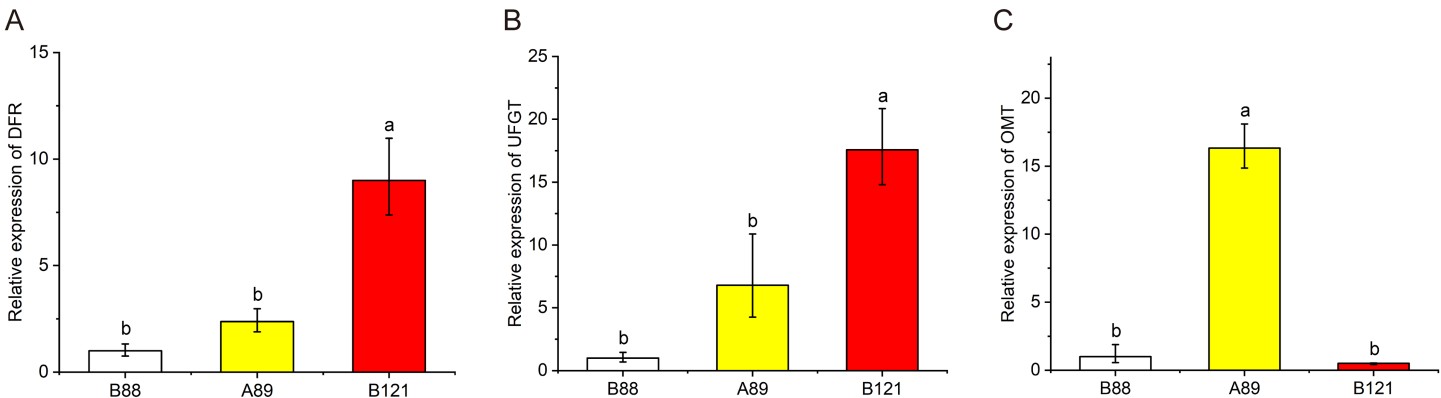

**Figure 6** **The expression profile of three structural genes (DFR, UFGT and OMT) in three *Nelumbo* cultivars (B88, A89, B121) was determined by qRT-PCR.**

## DISCUSSION

### Pigments determine the different colors of various sacred lotus cultivars

Since the mid-19th century, pigments have been extracted from colorful flowers to research the components, and a wide variety of pigments have been discovered, such as carotenoids and flavonoids. Carotenoids are considered to be the most widely distributed pigments in nature, which could not only be found in flowers, but also in fruits and storage organs in higher plants (*Zhu et al., 2010*). Previous study showed that carotenoids existed in petals of

different plant species, and contributed to the yellow color, such as in *Osmanthus fragrans*, butter yellow and golden yellow petals contain $\alpha$-carotene and $\beta$-carotene (*Han et al., 2013*). The carotenoids may have relationship with the yellow petals color of *N. lutea*, while the physical and chemical properties of carotenoids varied greatly with the flavonoids, which need deep research, separately. It is acknowledged that flavonoids are a large class of secondary metabolites, which also widely distributed in lotus (*Li et al., 2014a*). Previous studies have shown that anthocyanins belong to the red series of pigments and control flowers colors from pink to blue violet, while non-anthocyanin flavonoids belong to the pure yellow series, controlling colors from deep yellow to light yellow and approaching colorlessness (*He et al., 2011*; *Zhao et al., 2012*; *Zhao & Tao, 2015*). Cyanidin appears in red flowers, while delphinidin leans petals toward the blue spectrum (*Sun et al., 2009*; *Zhang et al., 2011a*). In tropical water lilies, the cultivars in which delphinidin 3-galactoside was detected presented an amaranth color, whereas those containing delphinidin 3′-galactoside appeared blue (*Zhu et al., 2012*). However, an overview of the relationship between color phenotype and chemical composition remains lacking in sacred lotus. In this study, 5 anthocyanins and 18 non-anthocyanin flavonoids were simultaneously detected, qualified and quantified in 207 sacred lotus cultivars, with four components firstly discovered. The composition and content of these anthocyanins and non-anthocyanin flavonoids were also investigated in 207 lotus varieties (Fig. 2), which can be help to screen the lotus germplasm with a high content of specific secondary metabolites. In addition, statistical analysis of flavonoids in lotus petals of different colors revealed that the contents of non-anthocyanins flavonoids were far higher than those of anthocyanins, and the distribution of these components differed significantly in lotus petals of different colors (Table 2). It can be seen that in lotus petals, quercetin derivatives are the most prominent flavonoids with the largest accumulation, which is consistent with lotus leaves (*Goo, Choi & Na, 2009*). And the content of kaempferol derivatives in lotus petals ranked second. What's more intresting is that anthocyanins are mainly distributed in purple-red and red lotus cultivars, and are rarely produced in yellow and white varieties. Among the five anthocyanins, malvidin 3-*O*-glucoside accumulated the highest in lotus petals, followed by delphinidin 3-*O*-glucoside. However, the petals are still red rather than blue, so the mechanism of lotus colors remains to be further studied. Unlike anthocyanins, no-anthocyanin flavonoids existed universally in all the lotus cultivars, and accumulated the highest in yellow petals. As anthocyanins and non-anthocyanin flavonoids were known to contribute a lot to the lotus colors (*Li et al., 2008*), significance analysis was conducted, and notable differences were observed in the contents of anthocyanins and non-anthocyanin flavonoids among cultivars of differing color. In addition, strong correlations were seen among color parameters, anthocyanin contents, and non-anthocyanin flavonoids contents (Fig. 3). The results further confirmed that there may be some intrinsic link between lotus color and pigments. Moreover, MLR analysis showed that there are many factors affecting petal color of sacred lotus. It is considered that TA is the essential factor responsible for sacred lotus color. With the increase in TA, a* value increased and L* value decreased, indicating the intense red and dark color of sacred lotus petals. In addition, Qc-3-Neo (**10**) was found to be the primary factor positively influencing the b* value, suggesting the higher Qc-3-Neo

(**10**) content, the deeper yellow petal color. Thus, it could be speculated that red color of sacred lotus petals due to anthocyanins content, while the yellow and white color are owing to non-anthocyanin flavonoids content (Fig. 4). While there are many studies using mathematical models to study flower color and composition (*Wang et al., 2001*; *Zhang et al., 2011a*), rarely studies have been done in this way in lotus. Our studies take advantage of large sample data to establish a mathematical relationship model between lotus color parameters and active ingredients, making it more digitized and visualized, which is of great significance to further research of the formation mechanism of lotus color.

## Regulatory genes contribute to flower coloration of lotus

In the present study, a model explaining the biosynthetic pathway of flavonoids in lotus was proposed to give us a better understanding of the connection between flower coloration and the modified patterns of anthocyanins and non-anthocyanin flavonoids (Fig. 5). The flavonoid biosynthesis pathway consists of five sub-pathways, indicating that the biosynthesis of flavonols is closely related to the biosynthesis of anthocyanins. Flavonols and anthocyanins appear to compete for a common substrate. Due to the different activities of flavonol synthase (FLS) and dihydroflavonol reductase (DFR), the same substrate, dihydroquercetin, produces quercetin (subpathway 1) and leucocyanidin (subpathway 2), respectively. In addition, it can be seen from the Pearson correlation coefficient that there is a certain negative correlation between Cy-3-Glc (**2**) and quercetin derivatives (Fig. 4), further verifying the competition between sub-pathways 1 and 2. The rule also applies to myricetin (sub-pathway 5) and leucodelphinidin (sub-pathway 4). More importantly, kaempferol derivatives were not significantly different among the four different colored lotus varieties (Table 2), possibly due to the lack of the corresponding competing anthocyanin pelargonidin. To date, many studies has reported that variation in regulatory genes is central to variation in pattern and intensity of pigmentation through the genetic basis of flower coloration (*Schwinn et al., 2006*; *Yamagishi et al., 2010*). *Sun et al. (2016)* identified and isolated several regulatory genes from sacred lotus, and a striking difference in *MYB5* gene was detected in two sacred lotus species through introducing *NnMYB5* into *Arabidopsis* plants, indicating *MYB5* is a functional transcription activator of anthocyanin synthesis, and related to the flower color difference between red flowers and yellow flowers. However, it still needs further effort to investigate the regulation mechanism of flavonoid biosynthesis in sacred lotus. Based on the biosynthetic pathway of flavonoids in lotus that we put forward, we further verified the expression of pathway genes in lotus petals of different colors (Fig. 6). The qRT-PCR results showed that the expression of DFR in red petals was significantly higher than that of yellow and white petals, which was consistent with high anthocyanin content in red petal cultivars. The expression levels of OMT genes in yellow petals were significantly higher than in red and white flowers, which was consistent with high non-anthocyanin flavonoids contents in yellow petal cultivars. Studies have shown that modulating gene expression in the flavonoid biosynthetic pathway can alter flower color (*Noriko et al., 2010*). Moreover, strategies to genetically engineer flower color through the flavonoid biosynthetic pathway have attracted widespread attention (*Nishihara & Nakatsuka, 2011*). The results suggested that essential

enzymes required for regulating flavonoid biosynthesis, such as DFR and OMT, can control the content of endogenous pigments in lotus petals. By manipulating specific lotus flavonoid pigments, alteration in flower color is expected. Previously, we sequenced these 207 sacred lotus cultivars. Combined with the metabolome data in this study, the regulatory patterns and metabolic pathways of flavonoids will be further studied and the relationship between the compositions of flavonoids and petal colors in sacred lotus can be hopefully explained at the molecular level.

## CONCLUSIONS

In this study, we developed an analytical method to detect a wide range of anthocyanins and non-anthocyanin flavonoids simultaneously in a dramatically shortened time period in the petals of 207 sacred lotus cultivars. Among the five anthocyanins and 18 non-anthocyanin flavonoids identified, four of the latter were reported for the first time in sacred lotus petals. Furthermore, the relationships between flower color and pigment composition and content were elucidated. The results showed that Mv-3-Glu (**5**) is the dominant anthocyanin, while Qc-3-Gln (**12**) accounts for most of non-anthocyanin flavonoids in sacred lotus cultivars. Moreover, there are significant differences in the anthocyanin and non-anthocyanin flavonoid contents among different cultivars, and MLR analysis confirmed that the TA was the most essential factor for determining petal color. A higher content of Qc-3-Neo (**10**) and Myr-3-GlcA (**6**) results in yellow flowers, while an increased TA and reduced Qc-3-Sam (**7**) content lend petals to turn red.

These results are indispensable to investigating the relationship between the compositions of anthocyanins, non-anthocyanin flavonoids, and petal colors in sacred lotus. The findings will contribute to our understanding of flavonoid biosynthesis, which may provide a theoretical basis for developing sacred lotus petals as a natural source of anthocyanins and non-anthocyanin flavonoids. In addition, this research will lay a solid foundation for subsequent investigations into metabolic and biosynthetic pathways in sacred lotus.

### Funding
This work was supported by the Scientific and Technological Innovation project of China Academy of Chinese Medical Sciences (grant number C12021A04515 and C12021A04108); National Natural Science Foundation of China (grant number 32170388); and the Fundamental Research Funds for the Central Public Welfare Research Institutes (grant number ZZ13-YQ-057), and National Key Research and Development Project (2019YFC1906601-01). The funders had no role in study design, data collection and analysis, decision to publish, or preparation of the manuscript.

### Grant Disclosures
The following grant information was disclosed by the authors:

Technological Innovation Project of China Academy of Chinese Medical Sciences: C12021A04515, C12021A04108.
National Natural Science Foundation of China: 32170388.
Fundamental Research Funds for the Central Public Welfare Research Institutes: ZZ13-YQ-057.
National Key Research and Development Project: 2019YFC1906601-01.

## Competing Interests

Author Gangqiang Dong was employed by the company Amway (China) Botanical R&D Centre. The remaining authors declare that they have no competing interests.

## Author Contributions

- Jing Liu performed the experiments, analyzed the data, prepared figures and/or tables, authored or reviewed drafts of the article, and approved the final draft.
- Yuetong Yu performed the experiments, prepared figures and/or tables, and approved the final draft.
- Gangqiang Dong analyzed the data, prepared figures and/or tables, and approved the final draft.
- Chenyang Hao analyzed the data, prepared figures and/or tables, and approved the final draft.
- Yan Liu conceived and designed the experiments, performed the computation work, authored or reviewed drafts of the article, and approved the final draft.
- Sha Chen conceived and designed the experiments, performed the computation work, authored or reviewed drafts of the article, and approved the final draft.

## Data Availability

The raw measurements are available in the Supplementary Files.

## Supplemental Information

Supplemental information for this article can be found online at http://dx.doi.org/10.7717/peerj-achem.22#supplemental-information.

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
