# Peer review of "Identification and quantification of flavonoids in 207 cultivated lotus (*Nelumbo nucifera*) and their contribution to different colors"

_PeerJ Analytical Chemistry, doi:10.7717/peerj-achem.22_

## Round 0.1 · original submission · Major Revisions

Dear authors, the reviewers have raised several critical points in your manuscript that require revision. Please pay special attention to the results and discussion section, especially concerning the ion fragments you discuss with other papers available in the literature. Also, English language requiring editing.

Reviewer 1 ·

Basic reporting

The manuscript 71194v1, titled “Identification and quantification of flavonoids in 207 cultivated lotus (Nelumbo nucifera) and their contribution to different colors,” details the flowers of N. nucifera content in flavonoids and relates it with their color. It is attractive to other researchers and within the journal scope. A few points need to be clarified before acceptance.
1. The authors can improve some English language. One example is the phrase in lines 15 to 17 that would be clearer if the authors used: This study performed a systematic qualitative and quantitative determination of five anthocyanins and 18 non-anthocyanin flavonoids from the petals of 207 lotus cultivars.
2. Please clarify if the full name of the species is Nelumbo nucifera Gaertn.

Experimental design

The manuscript 71194v1, titled “Identification and quantification of flavonoids in 207 cultivated lotus (Nelumbo nucifera) and their contribution to different colors,” details the flowers of N. nucifera content in flavonoids and relates it with their color. It is attractive to other researchers and within the journal scope. A few points need to be clarified before acceptance.
3. Explain the differences between your work presented and discussed in this manuscript and previous works cited between lines 39 to 50. In particular, the differences with the Deng et al. (2013) manuscript.
4. The authors should probably join the Results and Discussion part.

Validity of the findings

The manuscript 71194v1, titled “Identification and quantification of flavonoids in 207 cultivated lotus (Nelumbo nucifera) and their contribution to different colors,” details the flowers of N. nucifera content in flavonoids and relates it with their color. It is attractive to other researchers and within the journal scope. A few points need to be clarified before acceptance.
5. It would help the readers if table 1 included a column with references, and more ion fragments should be included. The accuracy used in the m/z values is probably less important than the indication of more ion fragments. For example, peak 8 identification is discussed in lines 183 to 187. It is indicated that the loss of m/z 146 and m/z 176 implies the linkage of the sugar unit to C-3. The reference supporting this information is Ablajan et al., 2006? The presence of m/z 447 in table 1 would help to understand.
6. The authors did not detect the common flavonoid nucleus fragmentation?
7. Table 2 is confusing. The abbreviations used for the compound’s names are indicated in Table 1, so there is no need to repeat them, and the footnote can be simplified. The footnote b should be for all results, not just for the first one. The table caption should probably indicate that each value is the mean ± standard deviation. Furthermore, it would be better if the notes and the letters used to identify differences differed. It will be less confusing for readers. Finally, are the differences within the column or the row?

Reviewer 2 ·

Basic reporting

No comment.

Experimental design

No comment.

Validity of the findings

No comment.

Additional comments

Dear authors,
The manuscript intituled “Identification and quantification of flavonoids in 207 cultivated lotus (Nelumbo nucifera) and their contribution to different colors” is interesting and suitable to publish. However, it needs a few adjustments.
Introduction
Your introduction needs more detail. I suggest that you improve the description at lines 35- 38 using the lotus as an object of the studies. A description/explanation regarding the main cultivars (can be the ones that you described in your study) can be included in lines 30-33. Also, highlight the hypothesis to provide more justification for the research.
Materials & Methods
Lines 94-97: Please, include at the end of these lines which section the authors will describe the total content of non-anthocyanin flavonoids (TF) and the total content of anthocyanins (TA).
Line 148: Please, edit the information regarding the “quick RNA Isolation Kit po(, Beijing, China)”. Also, line 150.

Results and Discussion sections
For me, these sections are the most critical point of the study. Therefore, I suggest a rearrangement and an improvement in these sections. For instance, some descriptions could be included in the discussion section and are in the results section (regarding table 2, for example). Moreover, the results of Figure 6 were not described in the results section. Besides that, the discussion is shown in the section “Putative flavonoid biosynthesis pathway of lotus” could be improved. Also, a test of significance can be included in Figure 6.

Lines 385-387: Why could it have happened?
Line 405: Please, edit this sentence.
Table 2. Please, standardize the significant letters.
Figure 1: Why these data were obtained from the mixture of three cultivars: “Qiaoshou-I” (71), “Jinlingningcui” (151), and “Silian13-I” (178)?
Figure 6: Please, correct the spelling of the title on this figure.

---

## Round 0.2 · Minor Revisions

You have made most of the required alterations to your manuscript. However, some details are still required. For example, Reviewer 1 indicated that a significance test must be included in Figure 6. Furthermore, a broader and more descriptive discussion is still required, further describing the compounds and the findings more. Furthermore, the color of the heat map in Figure 4 was changed, but the label is the same as described before and you did not describe this change in the response file.

Reviewer 1 ·

Basic reporting

Nothing to report.

Experimental design

Nothing to report.

Validity of the findings

Nothing to report.

Additional comments

The authors changed the manuscript accordingly to the previous comments and improved it significantly. So, the manuscript is suitable for publication.

Reviewer 2 ·

Basic reporting

The language was improved. They shared the raw data, but some columns are in Mandarin. I recommend that the authors translate everything into English.

Experimental design

As I described before in my last review, Figure 6 must be included a test of significance. I have included a comment in the PDF file regarding it.

Validity of the findings

The manuscript has new findings and can be accepted after revisions.

Additional comments

The manuscript is very interesting.
The authors accepted and made changes in this version but still need a broader discussion.
I recommend the authors consider describing the compounds and the findings more, so the manuscript can be more read and cited in the future.

My best regards,
The reviewer.

Annotated reviews are not available for download in order to protect the identity of reviewers who chose to remain anonymous.

---

## Round 0.3 · accepted · Accept

The authors have modified their manuscript according to the previous round of reviews, where only minor modifications were required. Thus, the paper is now acceptable for publication in PeerJ Analytical Chemistry.